# Gene Expression Profile in Different Age Groups and Its Association with Cognitive Function in Healthy Malay Adults in Malaysia

**DOI:** 10.3390/cells10071611

**Published:** 2021-06-27

**Authors:** Nur Fathiah Abdul Sani, Ahmad Imran Zaydi Amir Hamzah, Zulzikry Hafiz Abu Bakar, Yasmin Anum Mohd Yusof, Suzana Makpol, Wan Zurinah Wan Ngah, Hanafi Ahmad Damanhuri

**Affiliations:** 1Department of Biochemistry, Faculty of Medicine, Universiti Kebangsaan Malaysia Medical Center, Jalan Yaacob Latif, Cheras, Kuala Lumpur 56000, Malaysia; nurfathiah@ukm.edu.my (N.F.A.S.); ahmadimranzaydi@gmail.com (A.I.Z.A.H.); zulzikryhafiz@gmail.com (Z.H.A.B.); suzanamakpol@ppukm.ukm.edu.my (S.M.); wanzurinah@ppukm.ukm.edu.my (W.Z.W.N.); 2Faculty of Medicine and Defence Health, National Defence University of Malaysia, Kem Sungai Besi, Kuala Lumpur 57000, Malaysia; yasmin.anum@upnm.edu.my

**Keywords:** aging, gene expression, Malay adults, cognitive decline

## Abstract

The mechanism of cognitive aging at the molecular level is complex and not well understood. Growing evidence suggests that cognitive differences might also be caused by ethnicity. Thus, this study aims to determine the gene expression changes associated with age-related cognitive decline among Malay adults in Malaysia. A cross-sectional study was conducted on 160 healthy Malay subjects, aged between 28 and 79, and recruited around Selangor and Klang Valley, Malaysia. Gene expression analysis was performed using a HumanHT-12v4.0 Expression BeadChip microarray kit. The top 20 differentially expressed genes at *p* < 0.05 and fold change (FC) = 1.2 showed that PAFAH1B3, HIST1H1E, KCNA3, TM7SF2, RGS1, and TGFBRAP1 were regulated with increased age. The gene set analysis suggests that the Malay adult’s susceptibility to developing age-related cognitive decline might be due to the changes in gene expression patterns associated with inflammation, signal transduction, and metabolic pathway in the genetic network. It may, perhaps, have important implications for finding a biomarker for cognitive decline and offer molecular targets to achieve successful aging, mainly in the Malay population in Malaysia.

## 1. Introduction

A decline in cognitive function is the major hurdle to achieving successful aging. Malaysia is a multi-ethnic and developing country, made-up of several ethnic groups, in which Malays form 50.8% of the total population [1]. Indeed, Malaysia has made a remarkable demographic transition in the past decade. It is predicted to become an aging nation by 2050, when 15% of the population will be aged 65 and older [2]. With a rapidly growing aging population, cognitive impairment among the elderly will also increase exponentially in the coming decades. Our previous study revealed that about 15% of healthy Malay adults had cognitive decline, where cognitive domains for memory, learning, and attention skills start to deteriorate from the age of 30 [3].

Having intact cognitive capabilities is critical for older adults in maintaining functional independence [4]. Cognitive deterioration is manifested by a number of symptoms, such as memory loss, decreased ability to maintain focus, reduced problem-solving abilities, and impaired communication skills [5]. However, cognitive decline may occur in the absence of symptoms, called pre-clinical Alzheimer’s disease (AD), in a cognitively normal individual with AD neuropathology driven by the accumulation of amyloid beta (Aβ) plaque and neurofibrillary tangles (NFT) [6,7]. Some evidence reported that cognitive decline in healthy individuals might occur beginning from the late 20s, based on the regional brain volume [8,9], myelin integrity [10,11], accumulation of NFT [12,13], and concentration of brain metabolite [14,15]. Therefore, it is clearly shown that cognitive function deterioration might not be noticeable in early adulthood. Still, pathological changes are progressing at the biochemical and neuroanatomical levels.

Although age is a crucial factor affecting the rate of cognitive decline, multiple cross-sectional studies showed its progression is attributed to a range of factors, including genetics, psychological, disease-related, environmental, and lifestyle factors [16,17,18,19,20]. A study showed that global cognitive functioning among Malay, Chinese, and Indian was significantly different in less-educated elderly indivdiauls in Singapore [21]. Tan et al. (2003) conducted a community-based study that investigated Apolipoprotein E (*APOE*) polymorphism among three ethnic groups in Singapore found that Malays have the high frequency of the ε4 allele, associated with elevated serum cholesterol levels, by downregulation of the low-density lipoprotein (LDL) receptor, and decreased LDL clearance [22]. On the other hand, Wan et al. (2004) reported that Malay groups have a low frequency of allele ε2 compared to Indian groups, where individuals with at least one ε2 allele tend to have lower levels of total plasma cholesterol due to the reduced binding affinity for LDL receptors [23]. It is well documented that the high frequency of allele ε4 and low frequency of allele ε2 increase the risk of diseases associated with hypercholesterolemia and increase the risk of AD [24]. Several studies have also reported that a higher prevalence of cardiovascular disease in the Malay population increases susceptibility to mild cognitive decline and AD [25,26,27]. Therefore, it is interesting and noteworthy that an APOE polymorphism study showed a greater risk of Malays getting AD.

Several studies have identified pathological processes at the peripheral level associated with decreased cognitive function. A study by Janelidze et al. (2016) found that increased Aβ levels in the brain and blood plasma are associated with vascular disease that contributes to cognitive impairment [28]. Another study conducted in healthy elderly individuals showed that plasma Aβ was associated with a faster cognitive decline rate that might predict a transition to AD [29]. Using blood as a biomarker, one study found 18 potential proteins that might predict early AD by 2–6 years, with close to 90% accuracy [30], and about 133 genes were identified in Alzheimer’s patients with different expression patterns compared to healthy control subjects with 98% accuracy [31]. Moreover, a follow-up study conducted by Grünblatt et al. (2009) found five genes in peripheral blood mRNA of demented and non-demented subjects showed a significant correlation with a lower mini-mental state examination (MMSE) score [32].

RNA extracted from PBMCs had a higher abundance of gene expression and produced greater signals in microarray as compared to whole blood [33,34]. Therefore, measurement of gene expression in PBMCs may offer an effective biomarker for cognitive research, as the sample is easily accessible, less invasive, and inexpensive fluid for biomarker identification, allowing repeat sampling. PBMCs consist of cells, such as lymphocytes, monocytes, and macrophages, which play important roles in the immune system, and could exhibit inflammatory mechanisms, specifically compared to serum or plasma in the aging process. Moreover, using PBMCs as biomarkers in the development of cognitive decline has not been sufficiently investigated. Thus, in this work, we aimed to profile gene expression changes in PBMCs dedicated to elucidate the mechanism of age-related cognitive decline during aging in the Malay population in Malaysia.

## 2. Materials and Methods

### 2.1. Subjects Recruited

The data in this study were a part of the Toward Useful Aging (TUA) study, funded by the Long Term Research Grant Scheme (LRGS), Ministry of Higher Education Malaysia. The subject screening was performed from May 2013 to March 2015 in various locations around Klang Valley and Selangor. Subjects were eligible for the study if they were between the ages of 30 and 60, had no known physical or mental illness, were of the Malay race, and did not have more than 15 years of education (years of education is associated with cognitive performance and a risk factor for cognitive decline). Subjects were excluded if they were diagnosed with a psychiatric disorder or an untreatable chronic disease, such as cancer, kidney failure, coronary heart disease, or uncontrolled diabetes. Smokers and pregnant women were also excluded from this study. A total of 160 subjects were enrolled via the random sampling method. They were divided into four groups according to their age (*n* = 40); age intervals for group 30 was (28–34), group 40 was (35–45), group 50 was (47–54), and the group above 60 was (57–79). From the 160 subjects, 72 subjects were included for the microarray analysis. All subjects were informed of the details of the study, and their written consent was obtained before they enrolled in the study. The study’s protocol was reviewed and approved by the Research and Ethics Committee of Universiti Kebangsaan Malaysia Medical Centre (UKM 1.5.3.5/244/LRGS/BU/2012/UKM_UKM/K04).

### 2.2. Samples Collection and RNA Preparation

Peripheral blood samples (10 mL) were collected in an EDTA tube (BD, Franklin Lakes, NJ, USA) and processed within 4 h of procurement at room temperature. The samples were frozen at −20 °C until further process for RNA isolation. Briefly, blood was mixed with Lymphoprep buffer (Axis-Shield PoC, Oslo, Norway) to isolate peripheral blood mononuclear cells (PBMCs). Then, the mixture was centrifuged at 1500× *g* and 30 min at 4 °C. PBMCs formed a pellet after centrifugation, and the supernatant was removed. Then, the total RNA was extracted in the TRI-reagent (Molecular Research Center, Cincinnati, OH, USA), and the RNA was immediately stored at −80 °C until further processing. The total RNA isolation was carried out using the RNeasy kit, according to the manufacturer’s protocol (QIAGEN, Chatsworth, CA, USA). RNA concentration was quantified using NanoDrop 1000A (Thermo Scientific, Wilmington, DE, USA), while RNA quality was characterized using the Agilent 2100 Bioanalyzer and the RNA Nano Chip (Agilent technologies, Palo Alto, CA, USA). Only samples with a purity of 1.8 to 2.0 (A260/A280) and RNA Integrity Number (RIN) of 8.0 and above were selected for gene profiling. The microarray experiment was designed to compare gene expression profiles among four age groups: 30, 40, 50, and over 60. The total RNA isolation was performed for quantitative real-time PCR (qRT-PCR) validation.

### 2.3. RNA Amplification and Microarray Chip Hybridization

Gene expression profiling was performed using the HumanHT-12 v3 BeadChip expression kit (Illumina Inc., San Diego, CA, USA) containing 47,123 unique transcripts. A total of 200 ng of total RNA from each sample was labeled using the TargetAmp™ Nano Labeling Kit for Illumina^®^ Expression BeadChip^®^ (Epicentre Biotechnologies, Madison, WI, USA) to synthesize cDNA. Then, in vitro transcription was performed to generate, and labeled single-stranded RNA (cRNA) by incorporating biotin; the samples were purified using RNeasy^®^ MinElute^®^ Cleanup Kit (Qiagen, Hilden, Germany). A HumanHT-12 v3 BeadChip expression kit (Illumina Inc., San Diego, CA, USA) was used to hybridize the biotinylated cDNA samples at 58 °C for 17 h. The Gene Expression BeadChips were then stained with Cy3-streptavidin dye reagent (Thermo Fisher Scientific Inc., Waltham, MA, USA). They were scanned for signal detection using an Illumina iScan and the Bead Scan Software (Illumina Inc., San Diego, CA, USA).

### 2.4. Real-Time QRT-PCR Validation

The genes with a different fold changed levels, had the highest statistically significant expression were selected for validation to verify the age-related changes derived from the microarray data. Genes and forward/reverse primer used are presented in Table 1. The same RNA samples used in the microarray experiments were performed using the two-step quantitative real-time reverse transcription polymerase chain reaction (qRT-PCR) using QuantiNova™ Reverse Transcription and QuantiNova™ SYBR Green PCR (Qiagen Inc., Germantown, MD, USA). Briefly, 2000 ng of total RNA was reverse transcribed according to the manufacturer’s instructions. Each 20 µL aliquot contained 1 µL reverse transcriptase, 4 µL transcriptase reaction mix, and 15 µL of total RNA or water as the negative control. The reaction mix was incubated for 5 min at 25 °C, 20 min at 45 °C, and 5 min at 85 °C to obtain the cDNA template. The gene was then amplified with a 10 µL of reaction mix consisting of 5.5 µL of 2× QuantiNova SYBR reaction mix, primers, and cDNA template. Each sample was amplified in triplicate, and the results were normalized against glyceraldehyde phosphate dehydrogenase (GAPDH) as a reference gene. FC was determined by the delta-delta-Ct comparative method, using the average of Ct values after subtraction with a Ct value of GAPDH.

### 2.5. Statistical Analysis

The SPSS version 22.0 (IBM, Armonk, NY, USA) was used to analyze real-time QRT-PCR data. All data were expressed as mean ± standard deviation. The differences were tested by one-way ANOVA, and the significance level was set at *p* < 0.05 for all tests.

### 2.6. Statistical Analysis of Gene Expression Profiling

Raw images produced were imported to Illumina Genome Studio Software Suite to obtain normalized gene expression. The final report of the normalized data was transferred to a third-party software, Partek Genomic Suite version (Partek Inc., St. Louis, MO, USA), to perform gene expression profiling analysis. A principal component analysis (PCA) plot was generated as a quality control step, and the batch effect was removed as a source of variation. Hierarchical clustering was generated to visualize gene expression patterns. A three-way ANOVA with FC −1.2 to 1.2 and *p* < 0.05 with the Benjamini and Hochberg false discovery rate (FDR) was performed across all samples. The differential expression genes (DEGs) were exported to Microsoft Excel to simplify the analysis further. The analysis compared DEGs between the younger group (30 year old) and the older age groups (40, 50, and 60) in aging. A Venn diagram was generated to demonstrated DEGs present in all groups. The list of DEGs was subjected to analysis further using Pathway Studio to identify biological pathways that were over-presented.

## 3. Results

### 3.1. Demographic and Cognitive Performance of the Subjects

Demographic data and cognitive function tests were reported previously [3]. All biochemical parameters in the blood were in the normative range described by [35].

### 3.2. Sample Characteristics

All samples had good quality RNA without degradation, as shown by the integrity number of RNA, RIN > 7 (Table 2). There was no significant difference among groups in the RIN isolated from PBMC between each group.

### 3.3. Gene Expression Profiling and Differentially Expressed Gene in PBMC

#### 3.3.1. Principal Component Analysis

Data obtained from the microarray experiment went through sample quality control according to the principal component analysis (PCA) criteria and hierarchical cluster analysis. PCA is a statistical tool used to visualize unsupervised multidimensional data sets for observation of the sample variability. According to the PCA criterion, samples of the same experimental conditions are expected to be positioned with each other and grouped closer in the PCA plot. The result showed that samples from different age groups were clustered and distinguished (Figure 1 and Figure 2a–c). This result could suggest the good quality of samples used, and the samples have similar biological conditions within the same group.

#### 3.3.2. Hierarchical Cluster Analysis

The relationship between age groups was visualized using hierarchical cluster analysis by applying one-way ANOVA (*p* < 0.05) analysis across all ages. The unsupervised hierarchical cluster analysis involved un-classification of a sample into any group/experimental condition (Figure 3), whereas datasets in the supervised analysis were classified into specific groups/experimental conditions (Figure 4). Based on the observations of Figure 4, the datasets of age groups 30 and 40 were well separated, showing that there were differences in expression profiles between the age groups. However, the dataset of age group 50 presented clustered together with age group 60. Several genes in age group 50 had similar expression profiles with age 60, and these gene expressions may have similar biological functions.

#### 3.3.3. Differential Expression Genes (DEGs)

DEGs of each age group in normal aging was statistically generated by the Benjamini–Hochberg t-test with a false discovery rate (FDR), multi-gene correction at (*p* < 0.05) and fold change (FC) ± 1.2 (Table 3). There were about 2478–4366 genes that were altered with the progression of age. The most significant differences found in G60 > G30 might be because of the huge age differences. The upregulated genes were more pronounced in the age group 60 vs. 30 than in age 40 vs. 30. Overall, the number of upregulated vs. downregulated genes was relatively balanced in all age groups, although most of the DEGs in each age group were upregulated.

A Venn analysis was performed to determine the overlapping of the DEGs found within the pairwise comparison (Figure 5). The data demonstrated that about 405 DEGs overlapped between age groups, suggesting that common molecular mechanisms in the peripheral blood may exist during the aging process.

The observation of these top 20 genes unexpectedly revealed that most DEGs were downregulated in all age groups (Table 4). Among the genes, only five selected DEGs attracted our interest based on a previous study on aging and their role in cognitive function; *PAFAH1B3* (*p* = 1.82 × 10^−6^ PF4, FC = 1.58), *TM7SF2* (*p* = 4.93 × 10^−7^, FC = 1.50429), *RGS1* (*p* = 6.68 × 10^−3^, FC = 2.04), *KCNA3* (*p* = 2.50 × 10^−7^, FC = 1.88), and *TGFBRAP1* (*p* = 1.04 × 10^−7^, FC = −1.50426).

### 3.4. Pathway Analysis and Biological Process

To gain insight into which functional annotation is most affected, the DEGs were further analyzed at the pathway level using the Pathway Studio analysis tool. Four major pathways that were involved in aging were mapped; inflammation, metabolic, signal transduction, and nociception pathway. Enrichment analysis of differentially expressed genes in the age group 40 compared to age 30 revealed that platelet activation via G-protein-coupled receptors (GPCR) signaling, p38 MAPK/MAPK14 signaling, and omega-3-fatty acid metabolism appeared to be affected (Table 5). The gene set in age 50 compared to 30 showed significant biological process involved in IL-1 signaling, JNK/MAPK signaling, and TGF-beta signaling (Table 6). The most significant biological processes involved in age group 60 compared to 30 were omega-3-fatty acid metabolism, arachidonic acid metabolism, and JNK/MAPK Signaling (Table 7).

### 3.5. Gene Validation

To validate the age-related expression changes detected in the microarray analysis, qRT-PCR was conducted using selected genes from the top 20 most significant genes that related to aging and cognitive function. The microarray and qRT-PCR results were comparable for six genes in the age group 40 and 50 (Figure 6a–c). However, the expression of *PAFAH1B3* in the age group 60 did not show a similar pattern with the microarray data. The deviation noted as the *PAFAH1B3* was identified as being upregulated in the microarray, but was downregulated in age 60 by qRT-PCR analysis, demonstrating the influence of the staining dye bias, PCR primer, and microarray probe difference, efficiency of different transcriptase enzymes, and normalization procedure between these two methods [36].

## 4. Discussion

Cognitive decline is an inevitable part of aging; however, the onset of decline can be delayed. In a previous study, we reported that advanced age was associated with increased oxidative DNA damage and protein oxidation, leading to decreased cognitive performance among healthy Malay adults in Malaysia [3]. We postulated that progressive oxidative damage induced by excessive free radicals reduced antioxidant capacity and increased proinflammatory reaction over time. Therefore, in this current study, the gene expression changes observed in the PBMCs may be linked to increased oxidative stress during aging, which may play a vital role in developing cognitive deficits. Our result also shows that DEGs found were consistent with age-related changes in ion channel activity, immune system, and cholesterol/lipid metabolism. Lu et al. (2004) proposed a clock mechanism whereby accumulating age-related DNA damage could selectively alter promoter regions of age-regulated genes [37]. Moreover, at the same time, we observed that these DEGs were highly conserved in the biological processes that were important for maintaining cognitive function, which could be the target for oxidative damage.

### 4.1. Alteration Patterns of Gene Expression in Different Age Groups

A gene encoded potassium voltage-gated channel (*KCNA3*) was differentially downregulated across ages. *KCNA3* is involved in regulating neurotransmitter release [38], insulin secretion [39], neuronal excitability [40], immune response [41], apoptosis [42], and cell proliferation [43]. This ion channel gene is mainly expressed in the nervous and immune system, which alters the function or mutation, and is related to many age-related diseases (reviewed in [44]). In particular, *KCNA3* hyperpolarizes the cell membrane potential and promotes Ca^2+^ influx through the calcium release into the cytoplasm, increasing and stimulating diverse cells signaling (reviewed in [45]). Inactivating the function may affect presynaptic action potential, increase calcium influx and neurotransmitter release, impair neuron firing, and influence synaptic transmission (reviewed in [46]). It is congruent with the data from the animal model induced-sevoflurane to impair cognition, which reported downregulation of *KCNA3* at the brain hippocampus, suggesting the essential role of this gene in learning and memory [47]. Moreover, in diabetic rats with reduced insulin receptor kinase activity, the downregulated *KCNA3* expression was reported associated with memory loss [48]. Conversely, in microglial, *KCNA3* acts as a key regulator in neuroinflammation, whereby prolonging activated microglial may have a detrimental effect that contributes to neurodegeneration. It was reported that *KCNA3* is upregulated when neurons are exposed to the β-Amyloid peptide, the main component of the senile plaques observed in the brain of AD [49]. Moreover, the immunostaining study using human brain cortices showed higher expression of *KCNA3* in the cortical microglial of AD patients, strengthening the role of *KCNA3* in the pathogenesis of AD [50]. Thus, it was shown that the role of *KCNA3* in neurodegeneration is inconsistent, depending on the cell type.

Upregulation of the lipid mediators gene, *PAFAH1B3* (platelet-activating factor acetylhydrolase), was detected in this Malay population as age increased. Expression of *PAFAH1B3* was expressed primarily in the central nervous system [51], erythrocyte [52], and reproductive system [53]. This gene is known to possess a potent proinflammatory mediator in diverse physiological and pathological processes (reviewed in [54]). *PAFAH* acts by binding to the G-protein-coupled seven-transmembrane receptor, which activates second messenger systems [55], such as glycogen synthesis kinase (GSK-3β), which is involved in the process of phosphorylation of the microtubule-associated protein [56]. In addition, in the postmortem brains of AD patients, the activated GSK-3β inhibited Wnt signaling pathways, contributed to impaired learning and memory in hippocampal areas [57]. Moreover, high expression of *PAFAH* in central nervous systems, due to the phospholipase A_2_ (PLA_2_) and arachidonic acid (AA) release, may regulate inflammatory pathways, which subsequent leads to long-term neurologic deficits [51,58]. In addition, *PAFAH* promotes excitotoxicity by enhancing glutamate release [59] and long-term potentiation [60] and exert neurodegeneration. Various studies provided evidence that significant alteration in *PAFAH* expression might influence cognitive capabilities. For example, cognitive studies on patients with schizophrenia and bipolar disorder found that reduced prefrontal cognitive function was associated with genetic variation in *PAFAH*. The gene was reported to regulate neuronal migration, which might cause alterations to the cortical development and, subsequently, reduce GABAergic neurotransmission [61]. Meanwhile, other studies have found that increased plasma PAFAH levels in patients with CAD were associated with accelerated cognitive decline and suggested early developmental markers of AD during aging [62,63]. Interestingly, Ciabattoni et al. 2007 reported that activation of PAFAH in AD patient plasma is associated with an increase in lipid peroxidation caused by vitamin E deficiency [64]. Thus, these findings may suggest that an increase in *PAFAH* expression amplifies the inflammatory cascades and is implicated in neurodegeneration.

*TM7sf2* was reported to be involved in cholesterol biosynthesis, which plays a vital role in cell signaling and maintenance of cell structure in the body [65]. We found that higher expression of *TM7sf2* in age group 40 might be due to the high accumulation of ROS during aging. This view is supported by Belleza et al. 2013, who showed that *TM7sf2* was associated with the upregulation of NF-KB and TNPα in the cellular response during stressful conditions [66]. In addition, Graham et al. (2010) observed that upregulation of *TM7sf2* in response to excessive cholesterol levels exert toxicity effects by increasing oxidative stress, leading to lipid peroxidation [67]. Increased iron levels in liver tissue are also correlated with elevated cholesterol content, associated with increased expression of *TM7sf2* gene, leading to increased oxidative stress responses [67]. However, the downregulation of the *TM7sf2* gene was found in the astrocyte cell of AD mice, which influenced the reduction of neuron development and synaptic transmission [68,69]. However, to date, there are no reports that show the association of *TM7sf2* genes with cognitive decline during aging.

In the 50-year-old subjects, the G protein-coupled regulatory gene (*RGS*) exhibited increased expression compared to age 30. *RGS* negatively modulates GPCR, a mediator of signaling transduction pathways, such as cell proliferation, cell differentiation, plasma membrane transport, and embryonic development [70]. *RGS* genes were reported to modulate oxidative stress and longevity in the various models, such as Drosophila models [71], astrocytoma cells [72], and Aspergillus fumigatus [73]. The finding may be explained by Wu et al. 2017, who reported that decreased RGS1 expression in insulin signaling pathways regulates the daf-16 gene and increases the expression of sod-3 and mtl-1, which play an essential role in eliminating ROS levels and promote longevity [74]. Therefore, we suggest that the high accumulation of ROS level might explain the high expression of RGS1 found in this study, as age increases. Interestingly, the transcriptomic study conducted by Leandro et al., 2018, showed promising findings that increased expression of the *RGS1* gene in the PBMCs of AD patients has potential as a peripheral biomarker [75]. This supports the hypothesis that PBMCs express molecular changes that occur in the neurons of AD patients.

The growth factor beta 1 (*TGF-β1*) gene, an anti-inflammatory cytokine, was downregulated in the age group of 60 compared to age 30. *TGFB-1* participates in regulating cell growth, apoptosis, and tissue repair after injury [76,77]. Evidence suggests that the *TGF-β1* regulatory mechanism was impaired in the pathogenesis of AD, which was linked to the neuronal damage, leading to cognitive impairment [78,79,80]. Alteration in the TGFB pathway during aging causes changes in *TGFB-1* release, Smad 3 activation, and microglial response during neuroinflammation [81]. It was reported that a deficiency of *TGF-β1* in AD animal models was correlated with Aβ pathology and NFT formation [78,82]. In the brain, *TGF-β1* is secreted by astrocytes regulating the activation of microglial, reducing the release of inflammatory cytokines and increasing reactive species. Hence, impaired *TGF-β1* activation could reduce the capabilities of microglial during neuroinflammation and participating in Aβ clearance. Therefore, several studies found that TGFB-1 has potential as an anti-amyloidogenic agent by reducing Aβ, and inhibiting the formation of NFT by promoting activated microglia [83,84]. Moreover, it was found in a postmortem AD patient that the expression of mRNA *TGF-β1* was negatively correlated with the formation of NFT [85], suggesting impairment of *TGF-β1*/Smad signaling in tau pathology. Moreover, decreased plasma levels of TGFB-1 were documented in patients with AD in multiple studies [86,87].

### 4.2. Biological Processes and Gene Differentially Expressed in Age

The four outstanding pathways that emerged from this study were inflammation, signal transduction, metabolic, and nociception pathways. Our data demonstrate that, in the Malay population, inflammation pathway was predominant until age 50. This may be explained as an individual progress through adulthood, with a variety of factors driving the aging process, including oxidative damage and an unhealthy lifestyle. Various studies have shown that inflammation and oxidative stress as pathophysiological processes involve cognitive decline [88,89,90]. Thus, normal aging processes associated with increased inflammation and accumulation of ROS may result in immune deficiency and drive the rapid progression of neurodegenerative disorders, such as AD [91]. However, evidence of inflammation contributing to the decline of cognitive function in healthy individuals is limited and inconsistent, although many studies in the mouse model have suggested such deficits [92]. Moreover, according to our data, when aging becomes more apparent, the metabolic pathway may become the main contributor to disease related to the progressive decline in endocrine function, body composition changes, and metabolic syndrome. Therefore, it is crucial to identify the biological processes involved in different age groups for early cognitive impairment management.

#### 4.2.1. Platelet Activation through GPCR Signaling

We have shown that the inflammation pathway by platelet activation through GPCR signaling was affected in the age group of 40 compared to age 30. Upon cell injury, platelet agonists, such as platelet-activating factors (PAFs), secrete and bind to GPCR receptors, a family of membrane proteins with seven transmembrane domains that elicit intracellular signaling through heterotrimeric G protein [93]. As discussed earlier, PAF is a lipid second messenger in regulating inflammatory and apoptotic activity—a contributor to the neurodegenerative mechanisms associated with cognitive decline. The GPCR signaling pathway is an extremely complex pathway. It modulates diverse cellular response implicated with various pathological processes, such as obesity, type 2 Diabetes mellitus, cardiovascular, immunological disorders, infectious diseases, cancer, and neurodegenerative diseases (reviewed in [94]). In Alzheimer’s disease, GPCR is involved as a modulator of amyloid-beta generation through the modulation of α-, β-, and γ-secretases in the proteolysis of amyloid precursor protein (APP) [95].

Activation of platelets also induces the secretion of serotonin membrane receptors that modulate cell signaling. Serotonin receptor subtype 2A or *5-HTR2A* belongs to the GPCR family widely distributed in the central nervous system, and has an essential role in learning and cognition (reviewed in [96]). In line with the findings of this study, several studies show age-related serotonin decline leading to cognitive decline and dementia (reviewed in [97]). A postmortem study found that loss of *5-HTR2A* in the temporal lobe area, which is associated with short-term memory, is correlated with the rate of cognitive decline in AD patients [98]. Meanwhile, Lorke et al. 2006 found a decline in neurons expressing *5-HT2A* in the prefrontal cortex of AD patients [99]. The severity of cognitive impairment in Alzheimer’s patients has been reported to correlate with reduced 5-HT2A binding [100]. A decrease in *5-HT2A* expression is said to be directly proportional to the significant loss of neurons [101] and the formation of NFT in the AD brain (reviewed in [102]). Moreover, a cognitive study among healthy subjects reported the association of *HTR2A* gene variations in memory episodes [103].

#### 4.2.2. p38 MAPK/MAPK14 Signaling

The findings from this study found that P38 mitogen activated protein kinase (MAPK) activation, as age advances, may be involved in cognitive decline progress. P38 MAPK is activated in response to inflammatory cytokines and other stimuli, including hormones, G protein-coupled ligands, and ROS [104]. Postmortem brains of AD patients confirmed that p38 MAPK activation occurs in early AD progression [105,106]. A previous study revealed that Aβ-induced P38 activation leads to increased tau phosphorylation and promotes the amyloidogenic processing of APP [107]. In addition, an increase in p38 activity was reported to be due to ROS leading to loss synapse and aggravating cognitive function [108]. Interestingly, a clinical study has reported that p38 MAPK phosphorylation in Alzheimer’s patients’ blood is positively correlated with disease duration [109].

The *ASK1* gene is a member of the mitogen kinase activated protein kinase (MAPKKK or MAP3K) that activates the p38 pathway. Our data showed it was dysregulated in the age 40 group. *ASK1* acts as an early sensor of ROS accumulation and plays an important role in signal transduction for homeostasis maintenance against redox imbalance [110]. Regulation of this gene is not only limited to apoptosis; it is also involved in inflammation and senescence [111]. Accumulating evidence indicates that *ASK1* plays a direct role in the decline of cognitive function, especially in the pathogenesis of Alzheimer’s disease. For instance, Peel et al. (2004) found that activation of *ASK1* by Aβ42 protein leading to tau phosphorylation exacerbated memory impairment in AD pathology [112]. Moreover, Aβ activates *ASK1* through ROS and induces neuronal cell death [113]. The role of ROS and obesity contributing to the cognitive decline was studied by Toyama et al., 2015, showing that *ASK1* is involved in cognitive decline due to long-term high-fat diets through hypoperfusion caused by hypoxia-induced injury and tumor necrosis factor alpha (TNF-α) induction [114].

#### 4.2.3. Activation through IL-1 Signaling

Upregulation of gene interleukin-1 (*IL-1*) was observed in the age 50 group, and may be linked to increase IL signaling pathway activation. *IL-1* is expressed by various sites, MAPK, and nuclear factor kappa B (NF-κB) cascade [115]. Previous studies have found that older individuals showed higher IL-1 expressions [116], and in AD patients, a significant elevation of circulating IL-1β serum was displayed compared to the healthy [117]. Studies on animal models implicated in IL-1 and lipopolysaccharide (LPS) have shown a role of *IL-1* in decreasing specific cognitive functions, such as learning and spatial working memory [118]. These findings are also supported by several cognitive studies that report that *IL-1* activation is associated with cognitive function and found that IL-1 secretion also inhibits long-term potentiation (LTP) in the hippocampus [119]. Normal or low IL-1 levels have been suggested to positively affect memory performance, but this study is limited to the animal model [119].

In the IL-1 signaling pathway, *TNF-α* expression was found to increase in the age of 50. *TNF-α* is a proinflammatory cytokine involved in mediating inflammatory responses and cognitive decline. Increased levels of TNF-α in the brain and plasma were reported to be detected in Alzheimer’s patients. Moreover, expression of *TNF-α*, and other systemic inflammations, such as IL-1, TGFβ, and tau protein, have been associated with MCI progression to AD [120]. A study by Sudheimer et al., 2014, supported the observation that IL-1, TNF-α, and a combination with higher cortisol, were associated with reduced hippocampal volume in healthy older participants [121]. In many brain pathologies, the TNF level-α expression was higher, which induced neuronal loss via microglial activating.

Individuals aged 50 showed dysregulation of the interleukin kinase 4 (*IRAK-4*) receptor expression in the IL-1 biological process. Under inflammatory conditions, *IRAK4* can phosphorylate *IRAK1* and induce the production of the TNF receptor associated factor 6 (*TRAF6*) and TGF-β kinase 1 (*TAK1*). This signaling molecule causes *NF-κB* nuclear translocation from the cytoplasm to the nucleus, resulting in the production of inflammatory cytokines and chemokines (reviewed in, [122]). From our data, the expression IRAK-4, in response to inflammation, suggests its association with cognitive decline. Studies have shown that decreased expression of *IRAK-4* in mice injected with Aβ leads to a decrease in *TRAF6* and *NF-κB* levels, which reduces gliosis by Aβ and improves mouse learning and memory [123].

#### 4.2.4. TGF-β Signaling

Dysregulation of the TGF-β biological process was found in the age 50 group. TGF-β coordinates tissue homeostasis by regulating cytokines production, cell survival, and cell death through signal transduction, and deficit in this signaling closely relates to the inflammatory pathway and cognitive decline in AD (reviewed in [124]). In the animal model, low expression of TGF-β was associated with reduced neurogenesis and reduced response to novelty [125]. In agreement with the human study, decreased TGF-β may contribute to cognitive decline in depressed patients and Alzheimer’s patients [80]. Moreover, Alzheimer’s patients with moderate to severe NFT exhibit decreased expression of TGF-β mRNA in the superior temporal gyrus region; this deficit is closely related to NFT formation [85].

In the TGF-β signaling biological process, expression of cell division cell cycle 42 (*CDC42*) was found dysregulated in the age group of 50. *CDC42* is a member of the Rho GTPase family, regulating various signaling pathways, including tyrosine kinase receptors, heterotrimeric G-protein coupled receptors, cytokine receptors, integrins, and responses to physical and chemical stress (reviewed in [126]. In the senescent endothelial cells, *CDC42* induced upregulation of proinflammatory genes by the activation of the NF-κB pathway, contributing to chronic inflammation [127]. Moreover, other studies have shown that higher expression of *CDC42* is associated with higher mortality in the human blood cells [128,129]. Deregulation of *CDC42* was also reported in degenerative neuronal diseases, which act as signal transduction pathways controlling actin-microfilament organization in mediating neuronal survival, apoptosis, and dendritic growth [130]. *CDC42* activity was reported to increase in hippocampal neuron cells treated with Aβ42 [131]. The neuron cells of AD patients exhibited high expression of *CDC42* compared with age-matched controls [132]. Furthermore, cognitive impairment is closely correlated with synaptic loss, as reported in AD patients. Saraceno et al. 2018 reported that *CDC42* expression was high in the AD brains, which postulated the synaptic compensation process to respond to the synaptic deficit [133]. However, there is no information correlating *CDC42* expression in the blood to cognitive decline.

The *NF-κB* gene was found dysregulated in the age 50 group compared to the age 30 group. *NF-κB* plays an important role in the aging process as it is a key mediator of the immune response pathways, inflammation, apoptosis, and metabolism (reviewed in [134]). This is in line with the results of this study showing that expression IL-1 and TNF are also increased in this age group. Several studies have suggested that ROS might stimulate *NF-κB* expression, depending on cell type and pathogenesis of the disease ([135]. For example, lipid peroxidation inhibits *NF-κB* activity, resulting in increased neuronal death due to Aβ release [136]. The role of *NF-κB* subunits, such as *p65*, *p50,* and *c-Rel* in cognitive function has been widely reported. Cognitive impairment of cognitive memory has been found in rats with decreased expression of *c-Rel* [137] and decreased anxiety seen in p50-deficient mice [138]. Moreover, mice lacking p65 show deficits in spatial memory [139].

### 4.3. Omega-3-Fatty Acid Metabolism

Functional analysis identified that DEGs in age 60 is associated with metabolic function involving the metabolism of omega-3 long-chain polyunsaturated fatty acids (LC PUFA). Omega-3 fatty acids are composed of alpha-linolenic (ALA), eicosapentaenoic acid (EPA), and docosahexaenoic acid (DHA). EPA and DHA have been widely shown to have protective effects on neurons and improve cognitive function due to their modulatory effects on synaptic plasticity and neuroinflammation [140]. The findings from epidemiological studies showed that neurological disorders, such as AD and MCI, are associated with decreased LC PUFA levels [141]. Conquer et al. (2000) reported that cognitive decline with aging is associated with a decline in plasma DHA levels [142]. In contrast, studies from Milte et al. (2011) found that MCI patients exhibited lower EPA levels in erythrocyte membranes and higher arachidonic acid (AA) levels compared to healthy control subjects [143]. However, some clinical trial studies report inconsistent findings and do not present substantial evidence to support omega-3 PUFAs as a supplement for the prevention or treatment of cognitive decline in obese adults [144].

In this omega-3-fatty acid metabolism biological process, *FADS* genes exhibited decreased expression in the 60-year-old group. The use of LC PUFA in the body depends on desaturase enzyme activity in the metabolic pathway of fatty acids. Dietary fatty acids are converted by the enzymes Δ-5 desaturase (D5D) and Δ-6 desaturase (D6D) through the enzymatic desaturation, and the elongation process encoded by the *FADS1* and *FADS2* genes [145]. Several studies reported SNPs in *FAD* are associated with lower lipid levels in the blood [146], glucose levels [147], and cardiovascular diseases [148]. A study by Caspi et al. (2007) reported that genetic variants modulate the effects of breastfeeding on cognitive function in the *FADS* gene [149] and attention-deficit hyperactivity disorder [150]. However, studies on the effects of *FAD* on brain integrity and cognitive function in older people are limited. It is possible that decreased *FAD* expression in the age 60 group impaired the desaturase enzyme function, resulting in low ALA conversion to EPA and DHA, which may affect cognitive function performance. Dietary fatty acid requirements can likely be optimized according to FAD genetic profiles, to achieve optimum intelligence.

The arachidonate 5-lipoxygenase (*ALOX5*) gene is one of the key regulators of cholesterol metabolism [151]. It plays a role in accelerating the conversion of arachidonic acids to leukotrienes [152]. *ALOX5* expression is reported to increase in the central nervous system of Alzheimer’s patients and during the aging process. Notably, this increase was found in the hippocampus, the brain area most vulnerable to neurodegeneration, which has an important function in learning and memory formation [153]. Studies on AD transgenic mouse models prove that the *ALOX5* gene is likely to modulate amyloidogenesis in AD. This is demonstrated by the genetic disruption of this gene, resulting in a significant reduction in the amyloid plaque and suggested that this effect is mediated by modulation of the γ-secretase pathway [154]. However, studies on the modulation of *ALOX5* gene expression are still at an early stage, and the role of *ALOX5* in cognitive decline remains unclear.

### 4.4. Insulin Action

Several studies have reported that aging is accompanied by insulin resistance and altered glucose metabolism. Hence, metabolic syndrome, such as diabetes, is commonly observed among elderly adults [155]. Increased ROS levels or prolonged exposure to oxidative stress during aging disrupt insulin signaling by activating a series of signaling pathways, such as NF-κB, JNK/SAPK, and p38 MAPK [156,157].

In this study, expression of the insulin-binding growth factor gene (*IGFBP-2*) was increased in the age group of 60. Increased expression of *IGFBP-2* in individuals over age 60 may be associated with decreased insulin-like growth factor 1 (IGF-1) expression. Gockerman et al., 1995, reported a higher level of IGFBP-2 acts to suppress the biological effects of IGF-1 [158]. In the systemic circulation, the bioavailability and functions of IGF-l were mainly regulated by IGFBPs. However, the mechanism of interaction of IGF-1 activity by IGFBP-2 has not been fully elucidated.

Studies suggest that low IGF-1 contributing to cognitive decline may be due to nutrient deficiencies and deficient protein intake [159], which is commonly observed in older people [160]. A similar pattern of the result obtained in our previous work demonstrated that the age group 60 displayed significantly lower albumin concentration levels than other age groups [35]. Previous studies have shown that age and IGF-1 levels are correlated with processing speed as measured by the digit symbol test in healthy older men [161]. Lower levels of IGF-1 are associated with reduced processing speeds, but do not affect fluid intelligence and memory [162]. Furthermore, IGF-1 is directly associated with MMSE scores in older adults with cognitive impairment [163].

## 5. Conclusions

In conclusion, this study provides comprehensive blood transcriptomic profiling that define the gene expression changes in different age groups. We identified thousands of genes that show upregulation and downregulation of expressions enriched in inflammation, ion channel activity, signal transduction, and metabolism. We further showed that ROS accumulation during aging might activate gene sets involved in biological processes common to inflammation and the metabolic pathway in different age groups. The overall finding from this study is summarized in Figure 7. These biological processes are possibly the culprits for the vulnerability of cognitive decline during aging in the Malay population. Furthermore, it would be useful to perform a prospective study and combine blood measures across different modalities, such as proteins, metabolites, and gene expression, for further biomarker accuracy.

## Figures and Tables

**Figure 1 cells-10-01611-f001:**
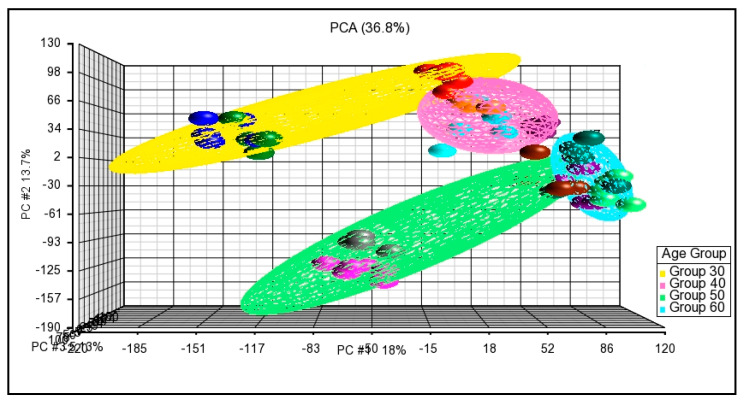
Principal component analysis (PCA) plot; age group 30 (yellow), age group 40 (pink), age group 50 (green), age group 60 (blue).

**Figure 2 cells-10-01611-f002:**
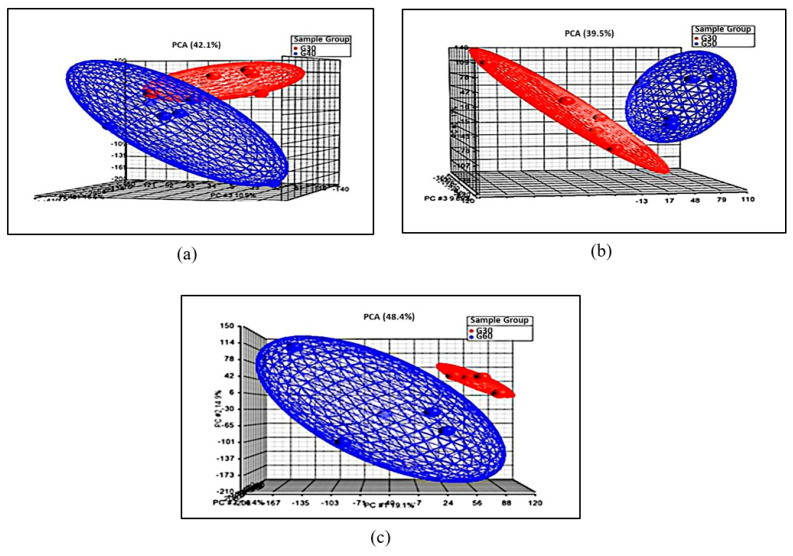
Principal component analysis (PCA). (**a**) Age group 40 (blue) compared to age group 30 (red), (**b**) age group 50 (blue) compared to age group 30 (red), (**c**) age group 60 (blue) compared to age group 30 (red).

**Figure 3 cells-10-01611-f003:**
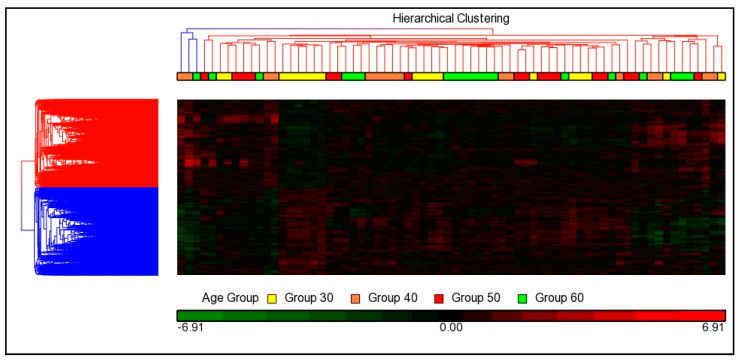
Unsupervised hierarchical cluster analysis of gene expression in age group; age 30 (yellow), age 40 (orange), age 50 (red), and age 60 (green).

**Figure 4 cells-10-01611-f004:**
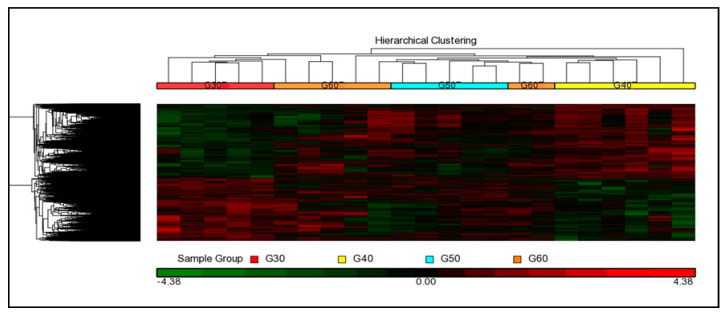
Supervised hierarchical cluster analysis of gene expression in normal aging age 30 (red), age 40 (yellow), age 50 (blue), and age 60 (orange).

**Figure 5 cells-10-01611-f005:**
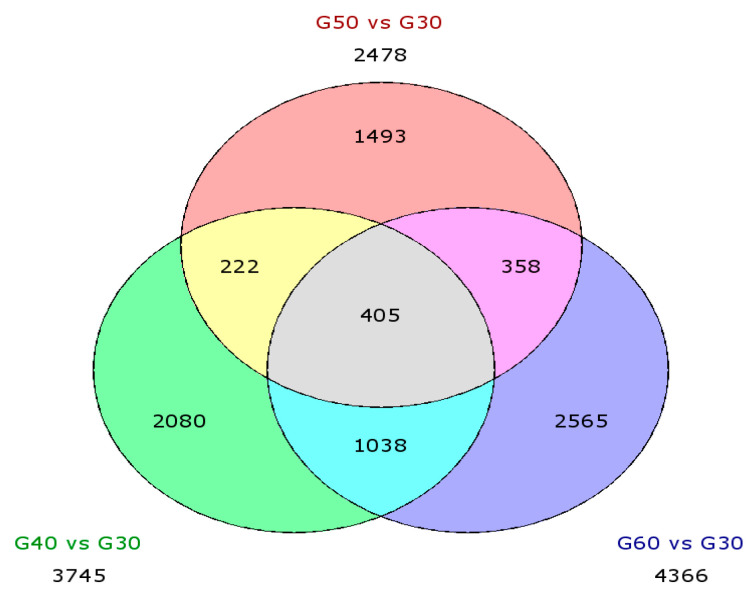
Venn diagram showed a significant overlap of differential expression genes in the pairwise comparison.

**Figure 6 cells-10-01611-f006:**
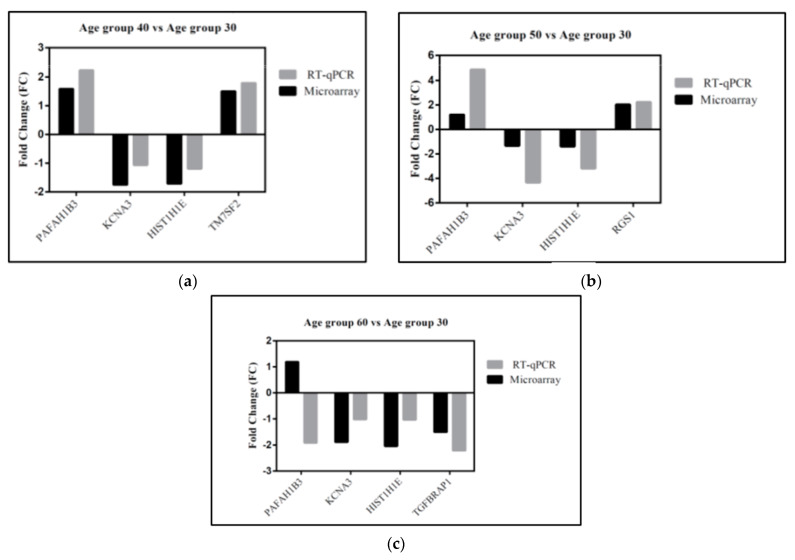
Quantitative validation of gene expression by qRT-PCR using selected genes from microarray experiment (**a**) age group 40 vs. age group 30, (**b**) age group 50 vs. age group 30, and (**c**) age group 60 vs. age group 30.

**Figure 7 cells-10-01611-f007:**
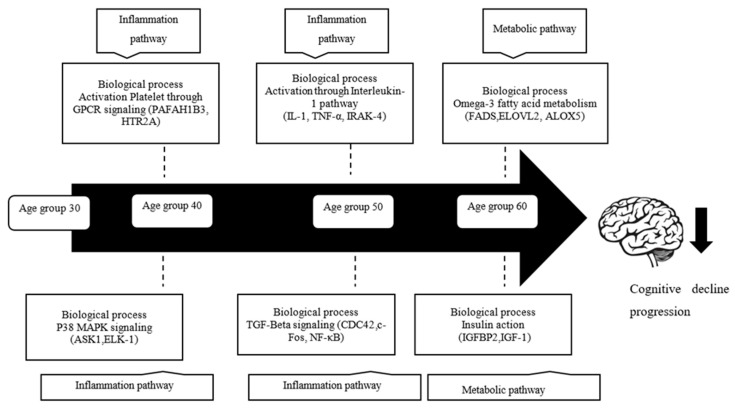
The summary on the profile of gene expression changes and biological processes associated with cognitive decline in age groups.

**Table 1 cells-10-01611-t001:** Primers used in real-time QRT-PCR analysis.

Gene Product	Forward Primer	Reverse Primer
*GAPDH*	TCCCTGAGCTGAACGGGAAG	GGAGGAGTGGGTGTCGCTGT
*KCNA3*	AAAACGGGCAATTCCACTGC	AACAAGGGCATAGGCAGACC
*HIST1H1E*	TTCCGGCTCGAATTGCTCTC	CTTCACGGGAGTCTTCTCGG
*PAFAH1B3*	GAATGGGGAGCTGGAACACA	CGCTCATTCACCAGTTGCAC
*TM7SF2*	GTCGCCTGCGCTATCCTATT	AGATGAAAGCGGTGAGGGTG
*RGS1*	TTGACTTCCGCACTCGAGAA	TGTTCACCCAGGGAGCCATA
*TGFBRAP1*	CTTCAAGAAGCCCGTGAACGA	ACATCTGGATGGTTCTGCGTT

**Table 2 cells-10-01611-t002:** RNA samples were used in the microarray experiment.

Group	Group 30	Group 40	Group 50	Group 60
RNA Integrity Number (RIN)	8.51 ± 0.58	8.33 ± 0.39	8.49 ± 0.47	7.63 ± 0.84

Data presented as mean ± SD, *p* < 0.05.

**Table 3 cells-10-01611-t003:** Number of dysregulated genes in different age groups as compared to age 30 at FC > 1.2, FDR *p* ≤ 0.05.

	G40 vs. G30	G50 vs. G30	G60 vs. G30
Up	1961	1318	2264
Down	1784	1160	2102
Total	3745	2478	4366

**Table 4 cells-10-01611-t004:** List of top 20 DEGs with FC sorted according to the highest *p*-value.

No	G40N vs. G30N			G50N vs. G30N			G60N vs. G30N		
	Gene Symbol	*p*-Value	FC	Gene Symbol	*p*-Value	FC	Gene Symbol	*p*-Value	FC
1	*LSM14B*	1.44 × 10^−7^	−1.68672	*LOC652537*	8.23 × 10^−8^	1.5223	*VPS13B*	1.19 × 10^−10^	−1.56519
2	*HS.574731*	1.52 × 10^−7^	−1.64622	*LSM14B*	2.08 × 10^−7^	−1.65683	*LSM14B*	2.55 × 10^−10^	−1.90272
3	*TM7SF2*	4.93 × 10^−7^	1.50429	*RNU4ATAC*	4.90 × 10^−7^	−2.66478	*SNRPD3*	9.60 × 10^−10^	−1.65141
4	*EIF3CL*	1.66 × 10^−6^	−1.5152	*EIF3CL*	1.46 × 10^−6^	−1.50767	*EIF3CL*	2.33 × 10^−8^	−1.62575
5	*PAFAH1B3*	1.82 × 10^−6^	1.5837	*LOC652455*	5.79 × 10^−6^	−1.56269	*SNORD95*	2.49 × 10^−8^	−1.77508
6	*SNORD13*	2.18 × 10^−6^	−2.2137	*TERC*	1.66 × 10^−5^	1.97178	*FLJ20309*	2.73 × 10^−8^	−1.65312
7	*BDP1*	3.79 × 10^−6^	−1.55882	*GPR183*	3.56 × 10^−5^	1.70917	*SNORD13*	3.59 × 10^−8^	−2.5244
8	*AAK1*	3.91 × 10^−6^	−1.69422	*INPP4B*	7.28 × 10^−5^	1.68441	*AAK1*	4.56 × 10^−8^	−1.87268
9	*TUBB4Q*	4.34 × 10^−6^	1.60485	*CTLA4*	8.09 × 10^−4^	1.69249	*TGFBRAP1*	1.04 × 10^−7^	−1.50426
10	*PHACTR2*	5.68 × 10^−6^	−1.58414	*HS.149244*	1.69 × 10^−3^	1.76622	*PAR5*	1.13 × 10^−7^	−1.55457
11	*KCNA3*	5.68 × 10^−6^	−1.75129	*HNRNPL*	3.47 × 10^−3^	−2.02885	*HS.438905*	1.78 × 10^−7^	−1.62903
12	*RNU4ATAC*	7.31 × 10^−6^	−2.38473	*SNORD13*	4.18 × 10^−3^	−1.55632	*KCNA3*	2.50 × 10^−7^	−1.88146
13	*DCUN1D1*	1.37 × 10^−5^	−1.5566	*KCTD12*	5.77 × 10^−3^	−1.55619	*BAGE5*	2.84 × 10^−7^	−1.58421
14	*FAM55C*	1.39 × 10^−5^	−1.6273	*RGS1*	6.68 × 10^−3^	2.04288	*SCARNA22*	5.89 × 10^−7^	−1.74235
15	*CPEB4*	1.56 × 10^−5^	−1.58899	*SIGLEC16*	8.41 × 10^−3^	−1.51751	*PAFAH1B3*	3.30 × 10^−6^	1.53754
16	*FAM82B*	2.28 × 10^−5^	−1.58899	*SNORA12*	8.41 × 10^−3^	−1.63572	*RNU4ATAC*	6.85 × 10^−7^	−2.58193
17	*UHMK1*	3.64 × 10^−5^	−1.63527	*FOS*	8.51 × 10^−3^	−1.64007	*HS.574731*	1.25 × 10^−6^	−1.54638
18	*RNF125*	3.72 × 10^−5^	−1.54084	*KIR2DL3*	9.25 × 10^−3^	−1.56381	*RNPC2*	2.43 × 10^−6^	−1.50256
19	*EP300*	6.04 × 10^−5^	−1.52404	*VNN1*	1.16 × 10^−2^	−1.53815	*FAM55C*	2.67 × 10^−6^	−1.67387
20	*RSBN1L*	7.21 × 10^−5^	−1.52066	*HIST1H2BG*	1.18 × 10^−2^	−1.57278	*ANKRD36B*	3.51 × 10^−6^	−1.55339

**Table 5 cells-10-01611-t005:** A list of statistically significant biological processes in age group 40 as compared to age group 30 (*p* < 0.05, FDR), sorted according to the *p*-value by Fisher’s exact test.

Biological Process	Overlapping Entities	*p*-Value	Hit Types
Platelet activation via GPCR signaling	*ARHGEF7*; *WAS*; *THPO*; *OC90*; *ERAS*; *ARPC2*; *HRAS*; *F2R*; *GNAI2*; *RAP1A*; *HTR2A*	1.63 × 10^−3^	Inflammation pathways
Branched chain amino acids metabolism	*PCCA*; *VARS*; *HSD17B10*; *ACAA1*; *HMGCS1*; *HADHA*; *HADHB*; *ECHS1*; *HMGCL*	4.62 × 10^−3^	Metabolic pathways
Inositol phosphate metabolism	*GDPD2*; *NUDT3*; *ITPKA*; *PI4K2A*; *PLCH2*; *ITPK1*; *ITPKC*; *IP6K3*	7.48 × 10^−3^	Metabolic pathways
p38 MAPK/MAPK14 signaling	*HIST1H3D*; *HIST1H3C*; *MAP3K5*; *MKNK1*; *EIF4A1*; *OC90*; *MAP3K11*; *HIST3H3*; *ELK1*	1.15 × 10^−2^	Signal transduction pathways
Ras-GAP regulation signaling	*DPYSL3*; *CASK*; *DOK2*; *HRAS*; *IL1RAPL1*; *ERAS*; *RAP1A*; *RASA2*	1.48 × 10^−2^	Signal transduction pathways
Respiratory chain and oxidative phosphorylation	*COX4I1*; *SDHB*; *ATP6V0E1*; *ATP6V1E1*; *ATP6V1F*	3.61 × 10^−2^	Metabolic pathways
CR3-mediated phagocytosis in neutrophils and macrophages	*MYO1G*; *C3*; *HRAS*; *ERAS*; *ARPC2*; *DES*	3.88 × 10^−2^	Inflammation pathways
Omega-3-fatty acid metabolism	*ALOX15*; *OC90*; *HSD17B10*; *ACAA1*; *HADHA*; *HADHB*; *ECHS1*; *CYP4F12*; *NACA2*; *FADS1*	3.9 × 10^−2^	Metabolic pathways
Leukotriene effect on vascular endothelial cell response	*PRKCG*; *MAP3K5*; *DOCK6*; *USE1*; *ADCY9*; *CDH1*; *BNIP1*	4.01 × 10^−2^	Inflammation pathways
synaptic endocytosis	*CLTA*; *ARRB1*; *SYT1*; *AP2B1*; *HRAS*; *ADCY9*; *ERAS*; *RAP1A*; *CALML6*	4.25 × 10^−2^	Nociception pathways
Neutrophil chemotaxis	*DOCK6*; *OC90*; *ERAS*; *ARPC2*; *NCF1*; *PF4*; *HRAS*; *ADCY9*; *GNAI2*	4.32 × 10^−2^	Inflammation pathways
Activation of complement cascade by pentraxins	*CFB*; *C8G*; *C8B*; *C3*; *C7*	4.66 × 10^−2^	Inflammation pathways
ERK/MAPK canonical signaling	*HIST1H3D*; *HIST1H3C*; *PRKCG*; *PLCH2*; *EIF4A1*; *OC90*; *ERAS*; *HIST3H3*; *MKNK1*; *HRAS*; *ADCY9*; *RAP1A*; *SPHK1*	5.48 × 10^−2^	Signal transduction pathways
CC chemokine receptor signaling	*CCR4*; *CCL19*; *PRKCG*; *LIMK1*; *WAS*; *TIAM1*; *ERAS*; *NCF1*; *HRAS*; *ADCY9*; *RAP1A*; *CCL20*	5.91 × 10^−2^	Inflammation pathways
Fatty acid oxidation	*HSD17B10*; *ACAA1*; *HADHA*; *HADHB*; *ATOX1*; *ECHS1*	6.82 × 10^−2^	Metabolic pathways
Riboflavin metabolism	*RFK*; *ACP2*; *DAK*	7.20 × 10^−2^	Metabolic pathways
Vascular endothelial cell activation by blood coagulation factors	*F10*; *PRKCG*; *OC90*; *ERAS*; *CALML6*; *HRAS*; *F2R*; *F2RL2*	7.24 × 10^−2^	Inflammation pathways
MC1R-related anti-inflammatory signaling	*HRAS*; *ADCY9*; *IL10*; *RAPGEF4*; *ERAS*; *RAP1A*	7.38 × 10^−2^	Inflammation pathways
Neutrophil recruitment and priming	*MAP3K5*; *OC90*; *ERAS*; *PF4*; *HRAS*; *GNAI2*; *CSF2*	7.73 × 10^−2^	Inflammation pathways
Notch signaling	*MIB1*; *NOTCH4*; *FBXW7*; *SLC35D2*; *CTBP1*; *ADAM17*	9.01 × 10^−2^	Signal transduction pathways

**Table 6 cells-10-01611-t006:** A list of statistically significant biological processes in the age group 50 as compared to age group 30 (*p* < 0.05, FDR), sorted according to the *p*-value by Fisher’s exact test.

Biological Process	Overlapping Entities	*p*-Value	Hit Types
Mast cell activation without degranulation through IL33/IL1RL1 signaling	*IL1RL1*; *IRAK4*; *NFKB1*; *MAP3K1*; *MAP3K7*	2.36 × 10^−4^	Inflammation pathways
Vitamin K metabolism	*F10*; *F7*; *PRRG4*; *VKORC1*; *VKORC1L1*	3.64 × 10^−4^	Metabolic pathways
JNK/MAPK signaling	*MAP3K1*; *CDC42*; *MAP3K7*; *TP53*; *DUSP5*; *MAP3K13*; *ELK1*; *FOS*	6.79 × 10^−4^	Signal transduction pathways
Overview of mast cell activation without degranulation	*IL1RL1*; *IRAK4*; *NFKB1*; *TLR9*; *MAP3K1*; *MAP3K7*; *CXCR4*; *CD180*; *FOS*	9.16 × 10^−4^	Inflammation pathways
Toll-like receptor-independent sterile inflammation	*IL1RL1*; *IRAK4*; *MAPK1*; *NFKB1*; *MAP3K1*; *CDC42*; *MAP3K7*; *EZR*; *FOS*	1.63 × 10^−3^	Inflammation pathways
Mast Cell Activation without degranulation through IL1R1 and TLR signaling	*IRAK4*; *NFKB1*; *TLR9*; *MAP3K1*; *MAP3K7*; *CD180*	3.49 × 10^−3^	Inflammation pathways
Tight junction regulation	*SDC3*; *CASK*; *SDC2*	4.02 × 10^−3^	Cell signaling
Prostaglandin E2 receptor signaling in neurons	*NFKB1*; *MAP3K7*; *GABRA6*; *GABRA5*; *GABRB3*; *TNFSF11*	1.20 × 10^−2^	Inflammation pathways
Neutrophil recruitment and priming	*MAPK1*; *MAP3K1*; *FOS*; *IRAK4*; *NFKB1*; *PF4*; *MAP3K7*; *TP53*	1.24 × 10^−2^	Inflammation pathways
Plasmin effects in inflammation	*MAPK1*; *MMP3*; *MAP3K1*; *CDC42*; *SERPINE1*; *FOS*; *NFKB1*; *PLCL1*; *CCL20*; *PLCE1*; *SPINK5*	1.95 × 10^−2^	Inflammation pathways
Function of macrophage M1 lineage	*IRAK4*; *MAPK1*; *NFKB1*; *TLR9*; *MAP3K1*; *MAP3K7*; *CD180*	2.16 × 10^−2^	Inflammation pathways
Irinotecan metabolism	*KL*; *CES2*; *CES1*	2.63 × 10^−2^	Metabolic pathways
Neutrophil activation via adherence on endothelial cells	*MAPK1*; *NFKB1*; *CD34*; *CDC42*; *SELPLG*; *NCF1*	2.99 × 10^−2^	Inflammation pathways
Mast cell activation without degranulation through tnfsf8 signaling	*NFKB1*; *MAP3K1*; *MAP3K7*	3.14 × 10^−2^	Inflammation pathways
Vascular endothelial cell activation by blood coagulation factors	*F10*; *F7*; *MAPK1*; *CTGF*; *FOS*; *IRAK4*; *NFKB1*; *GNA11*	3.15 × 10^−2^	Inflammation pathways
Inositol phosphate metabolism	*INPP4B*; *PLCL1*; *PIK3CA*; *INPP4A*; *PLCE1*	3.77 × 10^−2^	Metabolic pathways
Mevalonate pathway	*HMGCS1*; *GGPS1*; *IDI1*	4.25 × 10^−2^	Metabolic pathways
GABA(A) membrane hyperpolarization	*GABRA6*; *GABRA5*; *GABRB3*	4.68 × 10^−2^	Nociception pathways
TGF-beta signaling	*MAPK1*; *MAP3K1*; *CDC42*; *SERPINE1*; *ELK1*; *FOS*; *NFKB1*; *MAP3K7*; *ANAPC5*; *TGFBR1*	4.83 × 10^−2^	Signal transduction pathways
synaptic inhibition	*GLRA4*; *GLRA3*	4.94 × 10^−2^	Nociception pathways

**Table 7 cells-10-01611-t007:** A list of statistically significant biological processes in age group 60 as compared to age group 30 (*p* < 0.05, FDR), sorted according to the *p*-value by Fisher’s exact test.

Biological Process	Overlapping Entities	*p*-Value	Hit Types
Omega-3-fatty acid metabolism	*GGT1*; *CYP3A7*; *ELOVL2*; *GGT5*; *ALOX5*; *GGT7*; *TECR*; *ACOT7*; *CYP4A22*; *FADS1*; *CYP2D6*; *CYP2C18*; *CYP2A13*	2.13 × 10^−3^	Metabolic pathways
Arachidonic acid metabolism	*CYP2E1*; *AKR1C3*; *GGT1*; *CYP3A7*; *PTGDS*; *GGT5*; *ALOX5*; *GGT7*; *ACOT7*; *CYP4A11*; *CYP4A22*; *CYP2D6*; *CYP2C18*; *CYP2A13*	4.13 × 10^−3^	Metabolic pathways
JNK/MAPK signaling	*TRAF6*; *HSF1*; *CDC42*; *MAP3K3*; *DUSP26*; *DUSP9*; *ELK1*	1.20 × 10^−2^	Signal transduction pathways
Proplatelet maturation	*RASGRP1*; *JAK1*; *FLI1*; *ADCY10*; *RASGRF1*; *KRAS*; *CSF2RB*; *PF4*; *IL11*; *MKL1*; *CSF2*	1.49 × 10^−2^	Inflammation pathways
Neutrophil activation via adherence on endothelial cells	*SELL*; *CDC42*; *SELE*; *CR1*; *ACTR3*; *VAV1*; *ARPC2*; *NCF1*	1.54 × 10^−2^	Inflammation pathways
Neutrophil chemotaxis	*CXCL5*; *CDC42*; *VAV1*; *MYLPF*; *ARPC2*; *NCF1*; *ADCY10*; *ITGA4*; *KRAS*; *PF4*; *ACTR3*	1.95 × 10^−2^	Inflammation pathways
ERK5/MAPK7 signaling	*TRAF6*; *PML*; *MAP3K3*; *CTF1*; *NFE2L2*	2.43 × 10^−2^	Signal transduction pathways
Macrophage M2-related phagocytosis	*MYO1G*; *EPS15*; *KRAS*; *CDC42*; *VAV1*; *PTPRC*; *MYO1H*	3.05 × 10^−2^	Inflammation pathways
Omega-6-fatty acid metabolism	*CYP3A7*; *ELOVL2*; *TECR*; *ACOT7*; *CYP4A22*; *FADS1*; *CYP2D6*; *CYP2C18*; *CYP2A13*	6.27 × 10^−2^	Metabolic pathways
Insulin action	*FOXK1*; *DUSP22*; *FOXS1*; *PTPRF*; *DUSP9*; *ADCY10*; *ETV3*; *IGFBP2*; *KRAS*; *NR2C2*; *DUSP26*; *FOXN3*; *FOXN4*; *IRS2*; *FOXE1*; *FOXD4L3*; *FOXD4L1*; *SSH1*	6.34 × 10^−2^	Cell signaling
Glucose metabolism	*PFKM*; *ADPGK*; *ENO2*; *ALPL*; *PCK1*; *PGM3*	6.38 × 10^−2^	Metabolic pathways
Inositol phosphate metabolism	*INPP4B*; *PIK3CD*; *ITPKA; INPP4A*; *PLCB3*; *MTMR3*	6.38 × 10^−2^	Metabolic pathways
Caffeine metabolism	*CYP3A7*; *CYP4A22*; *CYP2D6*; *CYP2C18*; *CYP2A13*	6.88 × 10^−2^	Metabolic pathways
CR3-mediated phagocytosis in neutrophils and macrophages	*MYO1G*; *RASGRF1*; *KRAS*; *ACTR3*; *ARPC2*; *MYO1H*	7.23 × 10^−2^	Inflammation pathways
Adherens junction regulation	*DVL3*; *TCF7L1*; *CDC42*; *PTPRF*; *ZNF658*; *ZNF780B*; *SMAD5*; *BMP8A*; *HIPK3*; *TGFBR2*; *TEK*; *ZNF275*; *ZKSCAN5*; *NLK*; *WNT11*; *MAP3K3*; *MAPK4*; *ZNF3*; *ZNF563*; *ZNF461*; *HGF*; *ZNF418*; *ZNF550*; *RBAK*; *RHOQ*; *ARHGAP23*; *ARHGAP6*; *ARHGAP18*; *ARHGAP17*; *CRTAM*; *ROR2*; *ZNF274*; *IGFBP2*; *GDF1*; *PVR*; *ZNF417*; *ARHGAP22*	7.23 × 10^−2^	Cell signaling
Platelet Activation via GPCR Signaling	*CDC42*; *WAS*; *RASGRP1*; *ARPC2*; *PTGIR*; *RASGRF1*; *KRAS*; *ACTR3*	8.89 × 10^−2^	Inflammation pathways
Toll-like receptor-independent sterile inflammation	*TRAF6*; *BCL10*; *PRKCE*; *CDC42*; *HMGB1*; *IL1R1*; *KRAS*	9.07 × 10^−2^	Inflammation pathways
Glyoxylate and glycerate metabolism	*HAO1*; *GRHPR*; *GLYCTK*; *ENO2*	0.100737	Metabolic pathways

## Data Availability

Data available on request.

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
