# Peer review of "Gene Expression Profile in Different Age Groups and Its Association with Cognitive Function in Healthy Malay Adults in Malaysia"

_cells, 2021, doi:10.3390/cells10071611_

Round 1

Reviewer 1 Report

The article ‘Gene expression profile in different age groups and its association with cognitive function in healthy Malay adults in Malaysia’ by San et al. aims to determine the gene expression changes associated with age-related cognitive decline among Malay adults in Malaysia. The study is interesting and might provide valuable information regarding age-associated changes in gene expression in one ethnic group, however, there are some major concerns about the manuscript:

  1. The manuscript is written in a review format. The introduction and discussion parts are too long that it is really hard to understand what the authors are trying to convey. The authors need to significantly reduce these two sections (at least by 1/4th) and have to stay focused.
  2. The rationale for the study is not clear from the introduction. Also, the rationale for associating cognitive decline with changes in genes expression in PBMCs is not clear. These are the things that is to be added in the introduction.
  3. Is the presented data pooled data from males and females? Females have faster cognitive decline compared to males, therefore, data from males and females should be represented separately.
  4. In selecting study subjects, there is no statement about pre-diabetic conditions, obesity or BMI. These factors are critical because these are the conditions that can cause peripheral inflammation which in turn can contribute to central inflammation and cognitive decline.

Reviewer 2 Report

This study aims to evaluate the gene expression changes associated with age-related cognitive decline among Malay adults in Malaysia and provided a comprehensive blood transcriptomic profiling. Authors considered a list of genes that were enriched in inflammation, ion channel activity, signal transduction, and metabolism. Despite the importance of the findings, the pathway identified are commonly associated with the whole aging process (i.e ROS accumulation, insulin action or TGFb signalling) and could be considered as downstream effector rather than initial trigger of aging.  Probably, a methylation analysis rather than a transcriptome profile would have been useful to detect early age-related modifications (as https://doi.org/10.18632/aging.100861). In addition, protein assessment level in age group analyzed has not be performed. That would have helped to suggest different approaches in order to prevent/delay age-related cognitive decline. Furthermore, I think that would be really useful to suggest for each age group a specific therapeutic approach based on the paper results and literature. Finally, even if the complement system has not be investigated, it would be useful to add that Qiaoqiao Shi et al. demonstrate that C3-deficient mice were protected from the synapse, neuron loss, and cognitive decline typically observed in older mice, suggesting an important role of C3 in the aging brain. 10.3389/fimmu.2020.00734, 10.1523/JNEUROSCI.1698-15.2015. 

Minor

Please add figure tag in the results sections

Please specify “top 20 most significant genes “meaning in the paragraph 3.5

Round 2

Reviewer 1 Report

Concerns were addressed by the authors.